# Radio Coverage Assessment and Indoor Communication Enhancement in Hospitals: A Case Study at CHUCB [note 1]

**DOI:** 10.3390/s25164933

**Published:** 2025-08-09

**Authors:** Óscar Silva, Emanuel Bordalo Teixeira, Ana Corceiro, Antonio D. Reis, Fernando J. Velez

**Affiliations:** 1Instituto de Telecomunicações, Faculdade de Engenharia, Universidade da Beira Interior, Calçada Fonte do Lameiro, 6201-001 Covilhã, Portugal; oscar.silva@ubi.pt (Ó.S.); emanuelt@ubi.pt (E.B.T.); 2Departamento de Engenharia Eletromecânica, Faculdade de Engenharia, Universidade da Beira Interior, Calçada Fonte do Lameiro, 6201-001 Covilhã, Portugal; ana.corceiro@ubi.pt; 3Departamento de Física, Faculdade de Ciências; Universidade da Beira Interior, Rua Marquês d’Ávila e Bolama, 6201-001 Covilhã, Portugal; adreis@ubi.pt

**Keywords:** WBAN, wireless communications, mobile networks, ubiquitous cellular coverage, NB-IoT, radio spectrum measurements, ICT

## Abstract

**Highlights:**

**What are the main findings?**
Long Term Evolution showed better overall performance in terms of coverage and signal quality in the received power measurements conducted at CHUCB.Critical areas with poor coverage were identified, especially at points 17, 19, and 21, where there was an absence or weak signal quality in various technologies.

**What is the implication of the main finding?**
The identification of areas with poor coverage allowed for the proposal of targeted solutions, such as strategically setting up femtocells or DAS, to improve coverage and connectivity in critical areas.The improvement of cellular coverage is essential for efficient networking of health professionals, connected medical devices, and wireless patient monitoring systems.

**Abstract:**

The adoption of wireless medical technologies in hospital environments is often limited by cellular coverage issues, especially in indoor areas with complex structures. This study presents a detailed radio spectrum measurement campaign conducted at the Cova da Beira University Hospital Center (CHUCB), using the NARDA SRM-3006 and R&S^®^TSME6 equipment. The signal strength and quality of 5G NR, LTE, UMTS, and NB-IoT technologies were evaluated. Critical coverage gaps were identified, particularly at points 17, 19, and 21. Results revealed that operators MEO and NOS dominate coverage, with MEO providing better 5G NR coverage and NOS excelling in LTE signal quality. Based on the results, the localized installation of femtocells is proposed to improve coverage in these areas. The approach was designed to be scalable and replicable, with a planned application at Cumura Hospital (Guinea-Bissau), reinforcing the applicability of the solution in contexts with limited infrastructure. This work provides both technical and clinical contributions to achieving ubiquitous cellular coverage in healthcare settings.

## 1. Introduction

We live in an era witnessing unparalleled scientific and technological advancements in telecommunications, particularly in the development of wireless networking, leading to the emergence of solutions that transform social and professional interaction [1]. In the healthcare sector, wireless body area networks (WBANs) have emerged as an innovative solution for remote patient monitoring [2]. These networks use low-energy devices capable of wireless communication [3]. In general, these devices include sensors, actuators, and microcontrollers, positioned around or inside the human body, to monitor, collect, and transmit specific data, such as heart rate, blood oxygen levels, body temperature [3,4], blood pressure/24-h monitoring, and Holter. These data, along with clinical analysis (including blood tests) results, cardiac monitoring data/ECG, ultrasound outcomes, EEG, clinical imaging, or sleep apnea monitoring tests, among other ones, are then processed and diagnosed by healthcare professionals. This technological advancement, together with advanced telemonitoring of at-risk patients, is important because it allows healthcare professionals to monitor patients’ health status in real-time, thereby improving the provision of high-quality care [5].

Simultaneously, the increasing average life expectancy of human beings has led to a higher demand for hospital services, putting greater pressure on underlying human and financial resources. Many hospitalized patients present complex conditions, requiring constant monitoring to prevent serious health episodes. Studies show that warning signs, such as changes in heart rate and respiratory rate, can appear six to eight hours before critical events, such as cardiorespiratory arrests, being observed in 60% to 85% of cases. The early detection of these signs is essential to prevent worsening of the patient’s health condition and save lives.

Although current monitoring systems allow for the tracking of vital parameters in real-time, their effectiveness depends on the constant presence of healthcare professionals, which is often unfeasible due to the shortage of human resources. To overcome this challenge, the Cova da Beira University Hospital Center (CHUCB), through the Tele-monitoring of at-Risk Patients (TERI) project, implemented a telemonitoring system that combines preventive, continuous, and retrospective monitoring, optimizing available resources and ensuring faster and more effective responses. However, for the efficient operation of these technological solutions, a robust communication infrastructure is vital. The effectiveness of WBANs and system setups, like the ones from TERI, depends on reliable cellular coverage capable of ensuring the continuous and secure transmission of patient data.

Despite advances in information and communication technologies (ICTs) and their applications to healthcare, many hospitals still face significant challenges in ensuring suitable cellular and wireless local area networking coverage [6]. As demonstrated by Ahmed et al. [7], hospitals require minimum QoS thresholds that are not universally met due to the impact of buildings structural aspects in the signal propagation. Our study advances this field by empirically evaluating these metrics across multiple technologies (LTE, 5G NR, NB-IoT), complementing previous theoretical approaches. Previous studies indicate that the complexity of hospital structures and environmental interferences can compromise signal quality, leading to communication failures. These coverage gaps not only hinder the effective operation of WBANs but can also directly affect patient safety and well-being, especially in critical areas such as intensive care units and emergency rooms, since in the healthcare sector, data needs to be as complete as possible. Given the importance of ubiquitous communication, the effectiveness of networks in hospital environments fundamentally depends on the quality and availability of cellular coverage [8], as hospitals are very complex environments with various sources of interference and signal obstructions. Thus, it is essential to identify the areas that need improvement in cellular coverage to ensure the continuity and quality of service provision [9]. Recent research highlights the importance of wireless technologies, such as ZigBee, Wi-Fi, ultra-wideband (UWB), and Bluetooth/BLE (i.e., IEEE 802.25.4, 802.11, 802.3, and 802.1, respectively), in conjunction with mobile networks, e.g., narrow-band Internet of Things (NB-IoT), as well as the use of small cells, such as femtocells, in indoor environments, to improve coverage, as described by authors from [3,4].

Despite the advances observed in the integration of mobile technologies into healthcare systems, two gaps persist in the scientific literature regarding the evaluation of cellular coverage in hospital environments. The first pertains to the scarcity of empirical studies conducting comparative measurement campaigns involving multiple technologies—namely 5G, LTE, UMTS, and NB-IoT—within the same hospital setting. Most existing works adopt theoretical approaches or are limited to the isolated analysis of a single technology. Even the seminal study by Chen et al. [10], which describes the implementation of a standalone 5G network in a hospital in China, focuses exclusively on this technology, without addressing coexistence or comparison with other network layers.

The second gap relates to the near-total absence of applied studies in African contexts, particularly concerning the technical analysis of cellular coverage in healthcare facilities. Although investigations such as those by Bétila [11] and Achieng and Ogundaini [12] highlight the fragility of digital infrastructures and insufficient coverage as critical obstacles to the modernization of healthcare systems in Africa, these approaches are centered on macrostructural perspectives, without incorporating empirical measurements or multitechnological comparative analysis in clinical environments.

This study thus aims to contribute to bridging these two gaps: firstly, by conducting a multitechnological technical measurement campaign in a real hospital context; and secondly, by establishing a solid methodological foundation for the future replication of this study in a resource-limited environment, specifically at the Hospital of Cumura in Guinea-Bissau.

The aim of this study is to evaluate the quality of cellular coverage at the CHUCB, identifying areas where cellular coverage improvements are needed. This study will serve as the basis for future work to be carried out at the Cumura Hospital in Guinea-Bissau, where telecommunications infrastructures are significantly more limited. For this purpose, measurements of the radio spectrum were carried out at more than 20 points in the hospital, using measurement devices, such as the NARDA SRM-3006 spectrum analyzer and the R&S^®^TSME6 scanner, encompassing the sub-6 GHz bands, thus including the range of frequencies, up to 2.6 GHz, from this study [13]. Data obtained has allowed for a detailed analysis of coverage quality, enabling the proposal of handy solutions, such as the installation of femtocells or DAS in critical areas, to ensure ubiquitous, efficient, and seamless communication. Carrying out this study is essential to ensure that all sectors of the hospital have the required coverage to operate safely and effectively, thereby enhancing the quality of the healthcare provided to patients.

The remaining sections of the paper are organized as follows. Section 2 addresses the applied methodology, outlining the context and objectives of the measurement procedures and of the equipment used. In Section 3, the measurements conducted with the Narda SRM-3006 and the R&S^®^TSME6 scanner at the identified measurement points are addressed together with a comparison with cellular coverage maps in the region of CHUCB, followed by the analysis of the results in Section 4. In Section 5, the importance of ubiquitous cellular coverage in the support of easy-to-use telemedicine wireless devices in hospital environments is discussed, along with the discussion of a proposal for the deployment of femtocells or DAS and future expansion, and prospects for international expansion in developing countries. Finally, the conclusions are presented in Section 6 along with suggestions for further work.

## 2. Methodology

### 2.1. Context and Objectives

Measurements of the radioelectric spectrum were carried out at the Centro Hospitalar Universitário Cova da Beira (CHUCB) to evaluate the quality of the signal available in different areas of the hospital and its surroundings, as well as identifying zones with coverage deficiencies that could compromise the effectiveness of WBANs and other wireless communication-dependent systems [1]. This study was motivated by the need to explore possibilities for providing ubiquitous cellular coverage in the hospital and surrounding context, thus ensuring that all sectors of the hospital, especially the critical ones, have ubiquitous coverage that guarantees the continuity of operations and the safety of patients [13].

### 2.2. Radio Monitoring Equipment

For the measurement campaigns, two main measurement devices were used:The NARDA SRM-3006 (Narda Safety Test Solutions GmbH, Pfullingen, Germany) is a spectrum analyzer that allows selective frequency measurements for safety analysis and environmental measurements in high-frequency electromagnetic fields, with a range from 9 kHz to 6 GHz [1]. The complete SRM-3006 measurement system consists of a Basic Unit SRM-3006 and a three-axis antenna, which enables quick and simple isotropic measurements, with automatic determination of the three spatial components of the electromagnetic field, as illustrated in Figure 1 [13]. The NARDA SRM-3006 is widely used to monitor the coverage of cellular mobile networks and local wireless networks [1]. As signals in this frequency range of such magnitude are very difficult to sample digitally [13], this equipment combines analogue and digital signal processing to ensure absolute values, and limits high-frequency electromagnetic fields.

The R&S^®^TSME6 scanner (Rohde & Schwarz GmbH & Co. KG, Munich, Germany) is used for the analysis and optimization of mobile networks, as shown in Figure 2. This scanner, combined with the R&S^®^ROMES4 unit test software, version 21 (64-bit), can detect all available mobile network technologies across a frequency range from 9 kHz to 6 GHz, such as 5G NR, LTE, NB-IoT, and GSM, among others [1]. In addition to identifying the available mobile network technologies, the R&S^®^TSME6 can perform essential tasks involved in coverage measurements, such as detecting external interferences, measuring performance, and conducting quality analysis in mobile networks, e.g., bandwidth, total power received within the operator’s bandwidth [1,10].

**Figure 2 sensors-25-04933-f002:**
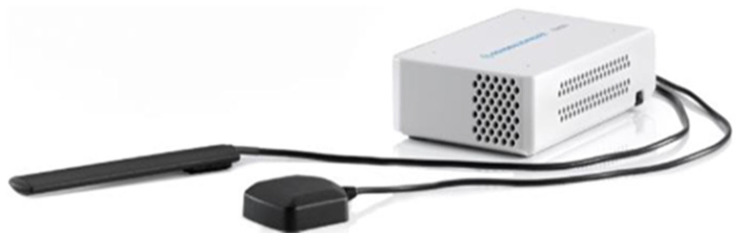
R&S^®^TSME6 drive test scanner.

The combination of the scanner with R&S^®^ROMES4 software allows viewing the available mobile networks, and which frequencies they are available on, while indicating the bandwidth and central frequency. The scanner supports various protocols such as 5G NR, GSM, LTE, NB-IoT, LTE-M, and TETRA. R&S^®^TSMA6 offers a detailed and comprehensive analysis of various parameters, performs an automated and reliable analysis, provides the view graphs of all detected technologies (with information to analyze various components of the cellular mobile network), with reduced energy consumption. The scanner combined with the R&S^®^ROMES4 unit test software also facilitates visualizing the timeline (with transmissions detected by the scanner) in graphical format. At the bottom of the homepage shown in Figure 3, in the lower horizontal bar, various menus with different functionalities are offered by the software: Welcome, Automatic Channel Detection (ACD), Navigation, 3GPP2 (CDMA200) Scanner, GSM Scanner, LTE Scanner, NB-IoT Scanner, and UMTS Scanner. Details are as follows: the Welcome menu lists all previously mentioned menus, while the ACD menu displays the frequency range and detected technologies, as shown in Figure 3.

The Navigation menu allows users to check the estimated route of data collection and estimate the geographic positions of base stations (with the indicated transmission directions), as shown in Figure 4. The 5G NR Scanner presents technologies detected in 5G NR, including received signal strength indicator, RSSI [dBm], graphs, and related parameters (e.g., Physical Cell ID). The LTE Scanner allows users to view detected technologies, power [dBm] graphs, and associated parameters.

The NB-IoT Scanner shows the detected operators’ networks in NB-IoT, including New Radio Signal Strength, NRSS [dBm], view graphs (through NRSSI), and related parameters. The 3GPP2 Scanner displays the operators (i.e., CDMA2000) at any point. The UMTS Scanner allows users to view detected UMTS operators, display received signal code power, RSCP [dBm], view graphs (providing information on which operator has the most reliable signal coverage), and associated parameters. Although the R&S^®^ROMES4 software displays TMN and Optimus as the names of the mobile operators, from now on, in this work, their current names, MEO and NOS, respectively, will be adopted.

### 2.3. Measurement Procedures

The measurements were conducted both inside and around the hospital, covering critical areas such as wards, the emergency department, imaging, the helipad, and parking lots (Figure 5). According to [14], the Hospital should provide ubiquitous access to mobile and wireless communications for the use of relevant applications and patient databases. It is recommended that the base stations be located closer to the users, thus reducing the level of electromagnetic fields. The farther the base station is from its users, the higher the power required to transmit; consequently, the higher the electromagnetic field strength.

Signal strength and connection quality were analyzed at each measurement point. Although the most used theoretical parameters in modern mobile networks include RSSI, RSRP, RSRQ, and SINR [1] in this study, the specific metrics available in the measurement tool by technology were considered: NRSSI for NB-IoT, RSCP for UMTS, Power for LTE, and RSSI for 5G NR. Measurements have been taken at different times of the day to capture temporal variations in signal quality.

The collected data were subsequently analyzed to identify the quality of coverage in each area, considering the following sectors of the Hospital:(i)Wards MEDICINE 1;(ii)Wards SURGERY 1;(iii)General emergency;(iv)Imaging;(v)Complementary exams.

The collected data were processed and compared to identify trends and patterns in the extension and quality of the mobile phone signal. Results were presented in graphs illustrating signal strength for different technologies at each measurement point, allowing for a clear visualization of areas where coverage is insufficient. Based on these results, specific recommendations were made for improving cellular coverage, including the proposal to install femtocells or DAS in critical areas.

## 3. Results

Although all the results have been obtained and analyzed, to maintain the conciseness of the manuscript, only the most relevant data are presented in the body of the text. The complete and detailed results of the measurements can be found in [13].

Measurements were undertaken in various areas inside and outside the CHUCB. For the measurements conducted with the NARDA SRM-3006, we present the results of six selected points for analysis. On the other hand, in the measurements taken with the R&S^®^TSME6 scanner, we analyze the worst measurement point for the four (4) technologies (LTE, UMTS, 5G NR, and NB-IoT), point 17. Furthermore, the summary with all six measurement points will be presented in Table 1. These points were carefully selected for analysis, considering their correspondence with the possible locations of the asset tracking system being designed to operate in the context of CHUCB.

The selected points are the same for both the NARDA SRM-3006 measurements and the scanner measurements. The R&S^®^ROMES4 unit test software, combined with the scanner, allowed us to record the time at each measurement point, which greatly facilitated the analysis of the results. Through the software, at each measurement point, it was possible to select a specific time to analyze the results. On the other hand, in the measurements undertaken with NARDA SRM-3006, each point has its corresponding measurement number, as presented in Table 1, thus facilitating the extraction of the data collected at each measurement point. Table 2 presents a listing of the frequency bands assigned to various operators for each available technology, which allowed for correlating the measurement results with the specific technologies and their respective operators [13].

Table 2 is essential for understanding the distribution of frequencies used by the main Portuguese operators, namely MEO, NOS, and Vodafone.

To facilitate the interpretation of the measurement results, typical thresholds for each technology and the considered metrics are shown in Table 3.

### 3.1. Measurement Results with the Narda SRM-3006

Graphs in Figure 6, Figure 7, Figure 8, Figure 9, Figure 10 and Figure 11 present the results of the measurements conducted with the NARDA SRM-3006 at the six selected points mentioned in Table 1.

The analysis of detected frequency bands in Figure 6 was made possible by providing a list of the frequency bands assigned to various mobile telecommunications operators, presented in Table 2.

Based on the data from Table 2, the following frequencies and their associated technologies were identified at point 13:A—760 MHz 5G NR F—1977 MHz UMTSB—819 MHz Technological Neutrality G—2132 MHz UMTSC—945 MHz GSM/UMTS/LTE H—2415 MHz Technological NeutralityD—1754 MHz GSM/UMTS/LTE I—2568 MHz Technological NeutralityE—1837 MHz GSM/UMTS/LTE J—2687 MHz Technological Neutrality

Similar lessons can be learned at the entrance to the outpatient consultations, which is not far from point 13. However, in point 13, other relevant technologies were identified, given that, due to the existence of a large, glazed surface next to the door, measurements were taken of technologies that are present only outside.

nPerf is a collaborative source of data on mobile network performance across different regions [15]. The nPerf platform gathers this data through crowdsourcing, by collecting geolocated signal strength and network performance information from mobile devices of end users who have installed the nPerf mobile application. Based on coverage maps obtained with nPerf, as shown in Figure 12, Figure 13 and Figure 14 (for MEO, NOS, and Vodafone, respectively), within the hospital, we consider cellular coverage maps for the region of CHUCB, which are complementary to the point-based measurements conducted by us.

As it was not possible to obtain coverage maps from September 2022, the exact period during which the indoor measurements were carried out, results for July 2025 are presented here.

### 3.2. Measurement Results with the R&S^®^TSME6 Scanner

The measurements for this study were conducted in January 2022. The R&S^®^TSME6 scanner, in combination with the R&S^®^ROMES4 software, provided a detailed and real-time analysis of the technologies available at each measurement point. The measurements recorded the exact moment each point was evaluated, allowing for a temporal analysis that complements the spatial assessment of coverage. Figure 15, Figure 16, Figure 17 and Figure 18 present the coverage graphs, representing the available received power for the four technologies detected in the measurements taken with the R&S^®^TSME6 scanner at measurement point 17 (mentioned in Table 1). These figures highlight specific characteristics that facilitate the interpretation and analysis of the data. The horizontal axis represents the signal peaks identified, indicating the different sub-measurement points within each main point. The vertical axis displays the corresponding unit values of the analyzed technology, reflecting the intensity of the measured signal in each case. To assist with visual interpretation, the signals are represented by three distinct colors: red, indicating a weak signal; pink, representing a medium signal; and yellow, which refers to a good or very good signal. Additionally, the analyzed operators are identified on the right side of the figures, highlighting the mobile networks measured at each point. It is important to note that, although the objective is to present multi-technology results, in some cases only one technology was detected or, in other cases, no technology was detected. Thus, in certain graphs, the representation may consist of a single bar or, in other cases, where no technology was detected, there are no bars, with each view graph having that meaning.

For each measurement point, the signal values (RSSI, Power, NRSSI, RSCP) were recorded for the 5G NR, LTE, NB-IoT, and UMTS technologies, enabling the immediate identification of the operator that offers the best signal quality for each technology. The meaning of the quantities is as follows:

**RSSI** (received signal strength indicator)—Indicates the strength of the received signal, in dBm, with ranges from −120 dBm (very weak signal) to 0 dBm (very strong signal).

**Power**—Refers to the signal strength, measured in dBm, usually between −30 dBm (very strong) and −110 dBm (weak).

**NRSSI** (NR signal strength indicator)—Indicates the signal strength, ranging from −140 dBm (very weak signal) to −40 dBm (strong signal).

**RSCP** (received signal code power)—Measures the average power of the received signal, in the range of −120 dBm (weak signal) to −40 dBm (strong signal).

Table 4 presents a summary of these best signal coverage values for each operator and different measurement points/technologies. A clear representation of the coverage of each of the four technologies (5G NR, LTE, NB-IoT, and UMTS) at different measurement points is provided, highlighting which operator (MEO, NOS, or Vodafone) provided the best coverage for each technology.

## 4. Analysis of the Cellular Coverage Results

The results of the measurements conducted with the NARDA SRM-3006 revealed important details about the distribution of cellular coverage at CHUCB. Several power peaks were identified at different frequencies, indicating the presence of some technologies, such as 5G NR and LTE, in various areas of the hospital. For example, the frequency of 760 MHz, associated with 5G NR technology, showed a significant peak, suggesting adequate coverage of this technology, in some of the considered measurement points. However, at higher frequencies, such as 2568 MHz, the coverage proved to be irregular.

Although the data obtained with the NARDA SRM-3006 allowed mapping some areas of the hospital that present significant coverage challenges, the level of detail was not sufficient. It became necessary to conduct additional measurements, with greater coverage, to more accurately determine the received power levels, frequencies, and available operators at CHUCB. Thus, new measurements were carried out using the R&S^®^TSME6 scanner, allowing for a more expedient identification of the operating frequencies and their respective operators.

Upon conducting a more detailed analysis of the measurement points, based on the new results, it was found that, at points 21, 23, and the entrance to the outpatient consultations, the 5G NR technology provided the best signal coverage. In contrast, at points 13, 17, and 19, it was the LTE technology that demonstrated better coverage. Overall, the LTE technology showed the best performance regarding power values and signal indicators (RSSI, NRSSI, RSCP). Moreover, both LTE and 5G NR technologies were present at all six measurement points analyzed, with LTE technology achieving the best coverage, except at the entrance to the outpatient consultations, where 5G NR showed superior coverage.

Regarding the operators, the results indicate that MEO and NOS were present at all six measurement points analyzed, while the Vodafone operator was absent at some of these points. In a comparative analysis between the two most frequently present operators (MEO and NOS), it was found that MEO provided better signal coverage at more points, for the four tested technologies (5G NR, LTE, NB-IoT, and UMTS).

Based on these results, for the analyzed technologies (LTE, UMTS, 5G NR, and NB-IoT), the zones requiring coverage improvement are mainly points 13, 17, and 21. These points recorded more frequent occurrences of weak signal, with points 17 and 21 showing the absence of some technologies, namely NB-IoT and UMTS at point 17, and UMTS at point 21. The analysis of results from this study reveals significant implications for hospital connectivity reliability. In multiple measurements, signal strength indicators for certain technologies fell below −85 dBm, which is beneath the generally recommended threshold for reliable communications in clinical environments. As emphasized by Ahmed et al. [7], hospital networks require consistent coverage to support proper operation of connected medical devices, while Niyato et al. [16] underscore how connectivity failures can compromise remote patient monitoring. These coverage gaps reinforce the need for targeted solutions such as the use of femtocells.

Although the data extracted from nPerf coverage maps do not fully represent the indoor environment of the hospital, they provide a complementary perspective on the evolution of cellular coverage in its surroundings. From the results obtained in 2025, MEO continues to have a considerably good coverage, with evidence of 5G coverage in the entrance to the outpatient consultations. For NOS, there is now evidence of good 5G coverage in the Emergency Service entrance, not well covered in our 2022 field tests. Vodafone also improved its coverage (4G, 4G+ in the entrance to the outpatient consultations), and clear evidence of good 5G coverage in the entrance of the Emergency Service.

## 5. Importance of Ubiquitous Cellular Coverage in Hospitals

Hospitals are dynamic and complex environments where seamless communication is essential for delivering high-quality patient care and ensuring operational efficiency. Cellular coverage plays a pivotal role in enabling the functionality of medical technologies, supporting telemedicine, and facilitating real-time data transmission. This section explores the critical importance of robust cellular coverage in hospital settings, highlighting its impact on medical systems, telemedicine, and proposed solutions for coverage improvement [17].

### 5.1. Impact on Medical Technologies

Modern hospitals increasingly rely on wireless systems to monitor patients, manage medical devices, and access electronic health records (EHRs). These technologies are indispensable for ensuring timely interventions and efficient workflows. However, their effectiveness is directly tied to the availability of reliable cellular coverage.

**Wireless Body Area Networks (WBANs)**—WBANs are a ground-breaking innovation that enables continuous monitoring of patients through wireless sensors. These sensors transmit vital signs such as heart rate, blood pressure, and oxygen levels in real-time, allowing healthcare professionals to respond quickly to changes in a patient’s condition [18,19,20]. For example, early detection of abnormal heart rhythms can prevent cardiac arrest. However, coverage gaps can lead to delays or interruptions in transmitting this critical data, potentially compromising patient safety and outcomes.

**Connected Medical Devices**—Many hospital devices, such as infusion pumps, ventilators, and vital signs monitors, rely on wireless networks to operate seamlessly [21,22]. These devices often transmit data to centralized systems for analysis and decision-making. Weak or inconsistent cellular signals can result in data transmission errors, delayed readings, or even device malfunctions. In critical areas such as intensive care units or operating rooms, such disruptions can have life-threatening consequences.

**Electronic Health Records (EHRs)**—The digitization of medical records has transformed healthcare by enabling doctors and nurses to access patient histories, test results, and prescriptions from anywhere within the hospital. This facilitates faster decision-making and improves the quality of care [23]. However, any failure in cellular coverage can hinder access to these records, delaying treatment and increasing the risk of medical errors.

In summary, robust cellular coverage is essential to ensure the uninterrupted operation of these technologies, which are integral to modern healthcare delivery.

### 5.2. Role in Telemedicine

Telemedicine has emerged as a transformative tool in healthcare, enabling remote consultations, patient monitoring, and real-time collaboration among specialists. Its adoption has accelerated in recent years, driven by the need to reduce patient flow in physical facilities and optimize hospital resources. However, the success of telemedicine depends heavily on reliable cellular coverage.

**Remote Consultations**—Telemedicine allows patients to consult with healthcare providers from the comfort of their homes, reducing the burden on hospital infrastructure. For example, patients with chronic conditions can receive regular check-ups without traveling to the hospital. Weak cellular coverage can disrupt video calls, cause delays in data sharing, or lead to dropped connections, undermining the effectiveness of remote consultations.

**Continuous Patient Monitoring**—Telemedicine enables healthcare professionals to monitor patients recovering from surgery or managing chronic illnesses. Devices such as wearable sensors transmit data to remote monitoring systems, allowing doctors to track patient progress in real-time. Poor coverage can result in gaps in monitoring, delaying interventions, and increasing the risk of complications.

**Collaborative Care**—Telemedicine facilitates real-time collaboration among specialists, even when they are in different locations. For instance, a surgeon in one hospital can consult with a radiologist in another during a complex procedure. Reliable cellular coverage ensures seamless communication and data sharing, enabling better decision-making and improved patient outcomes.

In areas with poor coverage, telemedicine services can be severely compromised, leading to delayed diagnoses, reduced access to care, and diminished patient satisfaction. Reliable connectivity is essential to ensure uninterrupted video calls and real-time data transmission [24], and can be severely compromised, leading to delayed diagnoses, reduced access to care, and diminished patient satisfaction. Addressing these coverage gaps is critical for hospitals aiming to expand their telemedicine capabilities.

### 5.3. Proposed Solutions for Coverage Improvement

To address the challenges posed by coverage gaps, hospitals can adopt targeted solutions such as femtocells and distributed antenna systems (DAS). These technologies are designed to enhance indoor cellular coverage, ensuring reliable connectivity in all areas within CHUCB.

**Femtocells**—Femtocells are small, low-power indoor base stations that provide localized cellular coverage. They are particularly effective in areas with high user density, such as wards, emergency rooms, and imaging departments. Femtocells can be deployed strategically to address coverage gaps without requiring extensive modifications to existing infrastructure. For example, installing femtocells in critical areas identified during the measurement campaign (e.g., points 13, 17, and 21) can significantly improve signal strength and reliability. In contrast to approaches focused exclusively on 5G networks, such as that proposed by Kumar et al. [25], our study advocates femtocell deployment as a more cost-effective and adaptable solution for hospital environments like CHUCB. Unlike 5G-only infrastructure, femtocells support multiple additional cellular technologies (e.g., LTE, UMTS, NB-IoT), aligning with the multi-technology signal environment observed in our measurements. Furthermore, this strategy is consistent with the recommendations of Singh et al. [17], who emphasize the importance of scalable and economically viable solutions for healthcare facilities operating under budget constraints. By enabling localized coverage across heterogeneous radio access technologies, femtocells provide a practical balance between service quality and deployment costs.

**Distributed Antenna Systems (DAS)**—DAS consists of multiple antennas distributed throughout a building, connected to a central base station. This system provides uniform coverage across large spaces, making it ideal for hospitals with complex layouts. While DAS is more expensive to install and maintain, it offers excellent scalability and can complement femtocells in areas with lower traffic density. Although their efficiency in terms of mobility and homogeneous coverage is recognized, their applicability in contexts with localized needs and limited budgets is less advantageous [26].

**Hybrid Solutions**—A combination of femtocells and DAS can be employed to optimize coverage. For instance, femtocells can be used in high-density areas, while DAS can cover transition zones or large open spaces [27,28]. This hybrid approach ensures cost-efficient and technically robust coverage tailored to the specific needs of the hospital.

The use of radiating cable, a technology traditionally applied in indoor coverage of specific environments, was also considered. However, in the particular context of the Centro Hospitalar Universitário Cova da Beira (CHUCB), its implementation faces substantial practical and operational barriers, compromising the feasibility of its adoption [29]. The installation of radiating cable presents considerable technical requirements, as it necessitates precise positioning to avoid contact with walls or metallic supports, which implies the use of specific support structures. The calibrated grooves along the cable, responsible for the gradual emission of the signal, must be appropriately oriented to prevent significant losses, making the installation process even more delicate. Additionally, this type of cable experiences progressive attenuation with increasing distance, requiring the use of amplifiers or segmentations [30]. This solution, nonetheless, proves particularly effective in linear environments, such as tunnels or long corridors [31]. Hence, despite its technical value in certain specialized scenarios, the application of radiating cable is not suitable for the punctual, dispersed, and localized coverage needs observed in the present study.

The comparative economic analysis conducted by [32] further supports this decision, demonstrating that in small to medium-sized buildings with high traffic demands, as is the case in hospitals, femtocells present a significantly lower total cost of ownership (TCO) compared to DAS systems. As long as the cost per FAP remains below EUR 440, femtocells are consistently the most economical solution. The advantage of DAS systems only becomes evident in very large buildings with low traffic density and when installation and infrastructure costs can be amortized over extensive areas [32].

Implementing these solutions will not only enhance connectivity but also support the growing demand for wireless medical systems and telemedicine services, ultimately improving patient care and operational efficiency in CHUCB.

### 5.4. Applications of NB-IoT in Sensor-Based Hospital Monitoring

Although the planned strategy for asset monitoring at CHUCB involves the use of Bluetooth Low Energy (BLE) tags integrated with more than 300 existing Aruba (Meridian) access points capable of integrating BLE and supporting BLE beacons, allowing real-time device location through internal georeferencing solutions. Among the software compatible with these BLE solutions, Aruba Meridian and Cisco DNA Spaces stand out, allowing real-time tracking of devices and people, with simple integration into the existing infrastructure. Narrowband Internet of Things (NB-IoT) emerges as a promising technology for hospital monitoring and tracking applications in contexts where there is no dense Wi-Fi infrastructure or effective BLE coverage. Although NB-IoT coverage at CHUCB has been limited, this technology offers technical advantages such as long range, efficient building penetration, and low energy consumption, making it suitable for devices that reliably report data sporadically [33,34,35]. However, as Ahad et al. [36] caution, implementing NB-IoT networks in hospital environments introduces critical security challenges, including vulnerabilities in data interception, medical device spoofing, and authentication protocols. These concerns become particularly relevant as our measurement results identify unreliable NB-IoT coverage at CHUCB’s points 17 and 21. In these areas, weak or absent signals not only impair communication reliability but also undermine the implementation of essential security measures. This dual challenge of connectivity and security underscores the necessity for future research addressing both coverage optimization and context-aware protection mechanisms for healthcare environments with constrained connectivity. The application of NB-IoT could be considered in outdoor areas, ambulances, or remote areas where such infrastructures are not available. This hybrid approach can extend the reach of tracking solutions, ensuring continuous coverage throughout the hospital perimeter.

There are already specific proposals in the literature for the use of NB-IoT for asset tracking in hospital environments, showing the feasibility of this approach [37]. Its integration with cloud-based IoT management platforms can enable functionalities such as real-time location, predictive analysis, and alert generation, with a direct impact on the efficiency of hospital services. Not only is asset management envisaged but also localization of patients will be explored (e.g., timely guiding patients to the most convenient service during their visit to the Hospital or localizing vulnerable patients).

Thus, although BLE continues to be the most promising technology in the hospital context, NB-IoT represents a relevant complementary technical solution, especially in future projects aimed at environments with limited indoor infrastructure or irregular coverage.

### 5.5. Future Expansion of the Study

The study conducted at the Centro Hospitalar Universitário Cova da Beira (CHUCB), as originally presented in [38], provides a solid foundation for future research and practical application of communication technologies in hospital environments, both locally and internationally. The expansion of this work will aim to address connectivity challenges in diverse healthcare settings, particularly in resource-limited environments.

A list of suggestions comprises the following aspects:1.**Asset Tracking Solution at CHUCB**—The study is exploring the implementation of a Bluetooth Low Energy (BLE)-based asset tracking system integrated with the hospital’s existing infrastructure. BLE tags will be paired with over 300 Aruba (Meridian) access points, which support BLE beacons and georeferencing solutions. The applications are as follows:(i)Medical Equipment Management—Real-time tracking of devices such as infusion pumps, ventilators, and monitors to optimize their usage and availability;(ii)Patient and Staff Tracking—Monitoring the location of patients and healthcare professionals to improve safety and operational efficiency;(iii)Dynamic Orientation—Guiding patients to the most convenient services during their hospital visit, enhancing their experience, and reducing delays.

This solution is expected to improve operational efficiency, safety, and the organization of workflows within CHUCB.
2.**International Expansion**—Aiming at targeting contexts in developing countries, the study will be extended to a case Study in the Hospital of Cumura in Guinea-Bissau, a reference unit for leprosy treatment in West Africa. Identified challenges are two-fold:(i)Connectivity Limitations—The hospital faces severe communication infrastructure challenges, including the absence of Internet access for healthcare professionals and manual recording of clinical data.(ii)Risks—These limitations pose significant risks to the quality of care provided, highlighting the need for accessible technological solutions.

In this context, the study identifies the need to investigate the feasibility of deploying low-cost and scalable technological interventions tailored to the local context. Specifically, the implementation of cellular coverage enhancement mechanisms, such as femtocells or distributed antenna systems (DAS), will be considered contingent upon the outcomes of a comprehensive radio spectrum measurement campaign to be conducted during the forthcoming phase of the research. Concurrently, the persistent reliance on paper-based medical records underscores the critical need to assess the potential transition to electronic health records (EHRs) as a means to improve clinical data accessibility, integrity, and continuity of care. Moreover, given the limited availability of local information technology infrastructure, the adoption of cloud-based platforms for clinical data management and decision support will also be evaluated, with particular attention to scalability, data security, and operational sustainability.

In the Hospital of Cumura in Guinea-Bissau, the same measurement and analysis approach (considered at CHUCB) will be applied, including radio spectrum assessments and cellular coverage evaluations. The study will compare the deployments at CHUCB (Portugal) and Cumura Hospital (Guinea-Bissau) to assess indoor communications enhancement and adapt solutions to local constraints, such as energy availability, reliability of network access provision, and financial limitations.

The ultimate goal is to develop a replicable technological model capable of enhancing internal communication, safeguarding clinical data, and supporting medical decision-making in resource-limited environments. Accordingly, the study will examine how solutions such as femtocells, BLE, and NB-IoT can be tailored to the specific technical and economic conditions of developing countries. The research to be conducted in Guinea-Bissau will thus serve as a natural and innovative extension of this work, with the potential to modernize hospital infrastructure in low-income settings, assess public policy and regulation enhancements, and help reduce global disparities in access to digital healthcare.

The relevance of this expansion is also demonstrated by recent studies addressing healthcare challenges across Africa. Bétila [11] demonstrated that the development of information and communication technologies (ICTs) significantly enhances the impact of public health investment, improving indicators such as life expectancy and infant mortality. However, the study also underscores that these benefits are contingent on the availability of reliable communication infrastructure, the focus of the interventions proposed here. Likewise, Achieng and Ogundaini [12] emphasize that fragile digital health infrastructure and poor data governance are critical barriers to implementing big data analytics for infectious disease surveillance in sub-Saharan Africa. Hence, improving cellular coverage for ICT availability enhancement in hospitals directly aligns with the recommendations of [11,12]. The Guinea-Bissau case study will benefit from incorporating these relevant lines of research.

Mobile network coverage maps for the area surrounding Cumura Hospital, based on publicly available data from the nPerf platform, are prospectively addressed in Figure 19 and Figure 20, even before performing the indoor coverage assessment. They illustrate the estimated signal coverage provided by the local MTN and Orange operators, respectively. Although these maps are not based on in situ measurements, they offer an overview of the existing coverage conditions in the region, and justify the relevance of this study.

Nevertheless, it is important to note that nPerf data does not distinguish indoor measurements. The apparent display of results within the building results from its geolocation accuracy, which is limited to approximately 50 m.

Hence, in situ measurements carried out in this work were essential to rigorously assessing signal quality. These maps will also serve as reference material for planning the next phase of fieldwork in Guinea-Bissau.

## 6. Conclusions

In this work, the cellular coverage at the Cova da Beira University Hospital Center (CHUCB) was evaluated. Based on the results obtained from the measurements carried out using the R&S^®^TSME6 scanner in combination with the R&S^®^ROMES4 software and comparing them with the previous measurements made with the NARDA SRM-3006, it was possible to achieve a greater level of detail regarding the distribution of different communications technologies in the hospital.

The analysis of the results allowed identifying specific areas that require coverage improvements. It was found that, at measurement points 21, 23, and at the entrance gate of external consultations, the 5G NR technology showed the best signal coverage. Whereas, at points 13, 17, and 19, it was the LTE technology that demonstrated superior coverage. The LTE and 5G NR technologies were present at all six measurement points analyzed, with LTE providing better signal coverage, except at the entrance to the outpatient consultations, where 5G NR showed superior coverage. In general, the LTE technology showed the best performance regarding power values and signal indices (RSSI, NRSSI, RSCP). It was found that, regarding the operators, the results indicated that MEO and NOS were present at all six measurement points analyzed, while the absence of the Vodafone operator was recorded at some of these points. MEO provided the best signal coverage in more locations and for the four technologies analyzed (5G NR, LTE, NB-IoT, and UMTS). The identified areas that require coverage improvements include points 13, 17, and 21. To address these shortcomings, the implementation of femtocells in areas with insufficient signal is proposed. This solution will optimize the quality of communication at CHUCB, ensuring robust and uninterrupted coverage, essential for the proper functioning of hospital systems, especially in critical areas.

Implementing these improvements will have a significant impact on the efficiency of hospital services, particularly in supporting health monitoring systems and wireless communication networks, facilitating telemonitoring of patients at risk, and ensuring that mobile technologies can be fully utilized to provide high-quality and safe healthcare. As emphasized by Ahad et al. [36], network security must form a cornerstone of healthcare’s digital transformation. In future extensions of this study, particularly at Cumura Hospital (Guinea-Bissau), where infrastructure limitations may increase risks, such as sensitive data interception, it will be essential to implement robust security protocols aligned with these authors’ recommendations. The confidentiality and integrity of medical data transmitted via mobile networks thus become paramount for ensuring reliable telemonitoring solutions in resource-constrained environments.

In addition to the proposed improvements in cellular coverage, the relevance of emerging solutions for real-time tracking within the hospital is also highlighted in this study. It is expected that the tracking system based on BLE and Access Points, currently under evaluation at CHUCB, will not be limited to the management of medical equipment and devices, but will also allow for the real-time location of patients. This functionality opens up new possibilities in terms of dynamic orientation of patients during their hospital journey and monitoring of vulnerable patients, contributing to enhanced safety and efficiency in the hospital environment.

In addition to the conclusions obtained based on the measurements carried out at CHUCB, this work aims to serve as a basis for a future international extension of the study to be conducted in a hospital in Guinea-Bissau. The choice of this new location is driven by the urgent need to understand the challenges associated with cellular coverage enhancement in hospital environments of developing countries, where telecommunications infrastructures are significantly more limited. The same measurement and analysis methodology will be applied in order to compare connectivity realities, identify critical areas, and propose viable solutions adjusted to local technical and economic constraints. This expansion of the study will not only validate the applicability of the proposed solutions but also contribute to the development of more accessible and scalable strategies to improve healthcare in contexts with limited resources.

nPerf is a valuable tool for providing a broad overview of cellular coverage, especially in outdoor environments, as demonstrated by recent publicly available data gathering in 2025. Its effectiveness is maximized when used in conjunction with detailed, real-time measurements for critical indoor areas, as demonstrated in the CHUCB field tests.

## Figures and Tables

**Figure 1 sensors-25-04933-f001:**
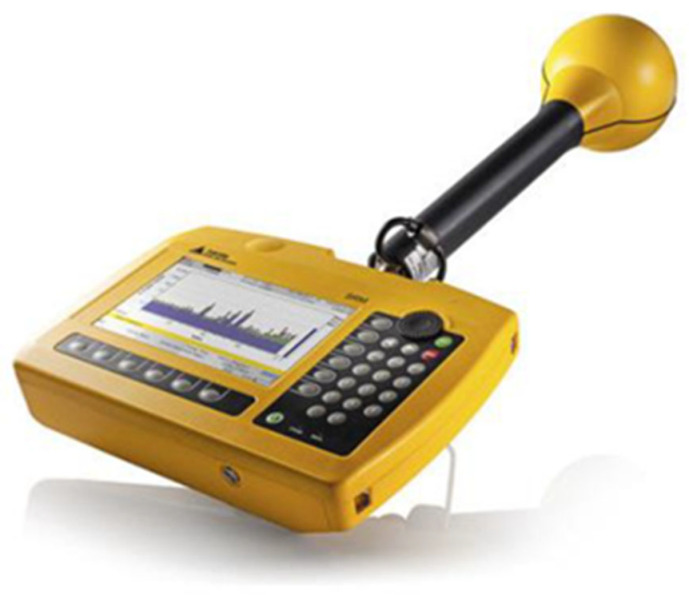
Narda SRM-3006 spectrum analyzer.

**Figure 3 sensors-25-04933-f003:**
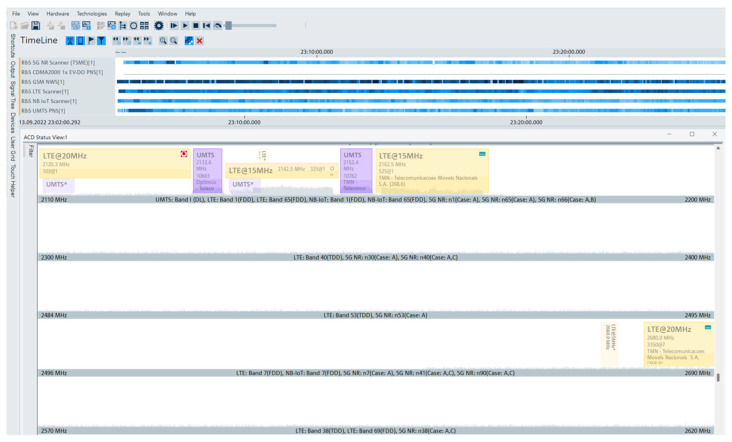
View of the automatic channel detection menu.

**Figure 4 sensors-25-04933-f004:**
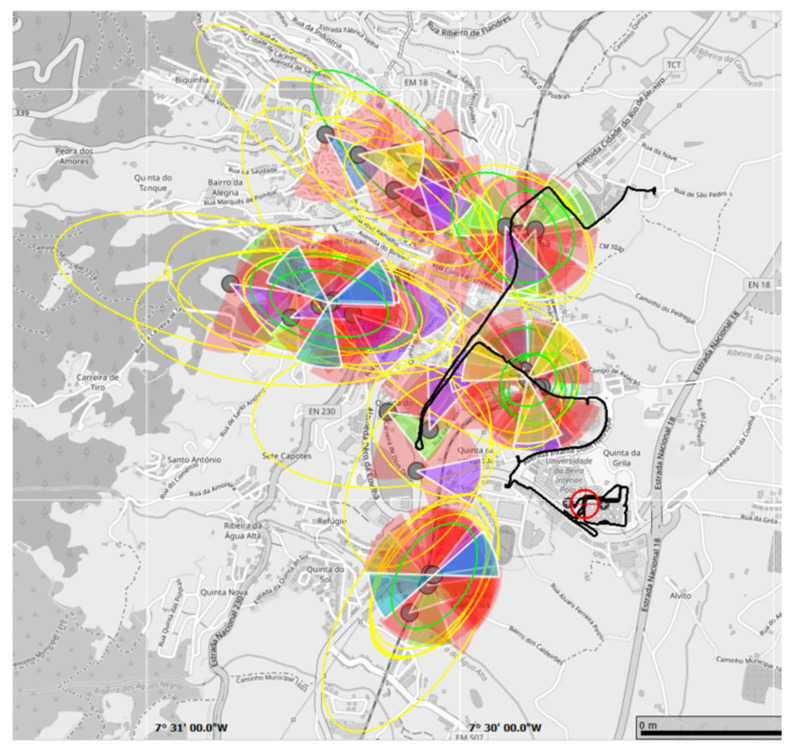
Navigation map obtained during the measurement campaign using the R&S^®^TSME6 scanner and ROMES4 software. Each coloured shape represents a detected cell, and the triangular segments within them indicate the antenna sectors or beams. The colours are automatically generated by the software to distinguish between sectors and do not reflect signal quality, frequency, technology, or operator.

**Figure 5 sensors-25-04933-f005:**
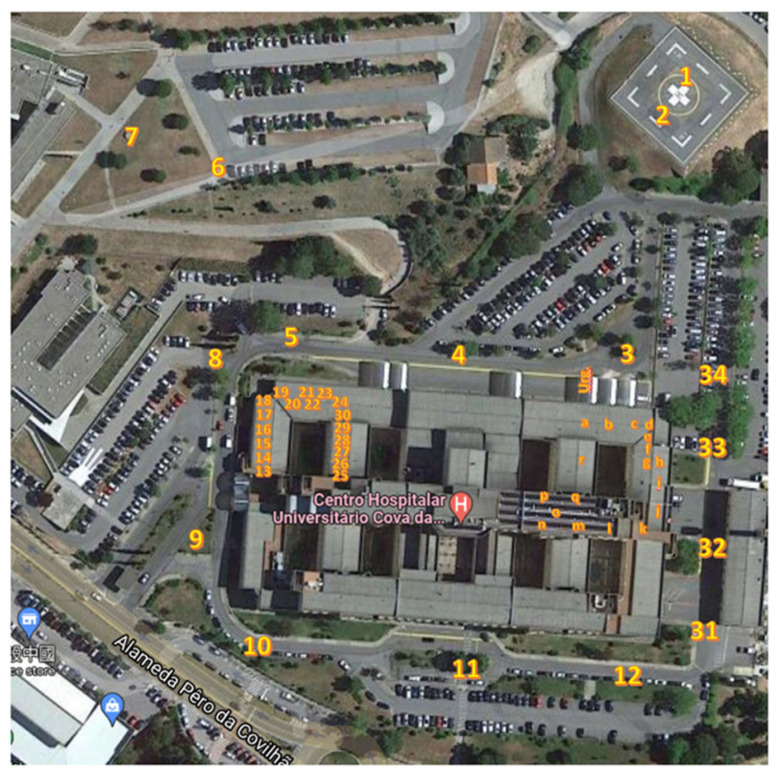
Map of measurement points at the Centro Hospitalar Universitário Cova da Beira. The numbered (1–34) and lettered (a–p) labels indicate spatial locations for possible measurements.

**Figure 6 sensors-25-04933-f006:**
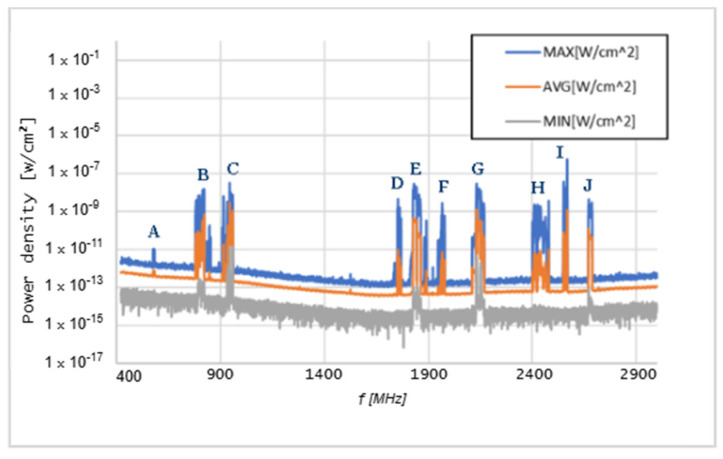
Graph of the measurement performed at point 13. Letters A–J indicate the detected frequency bands, which correspond to the allocations presented in Table 2.

**Figure 7 sensors-25-04933-f007:**
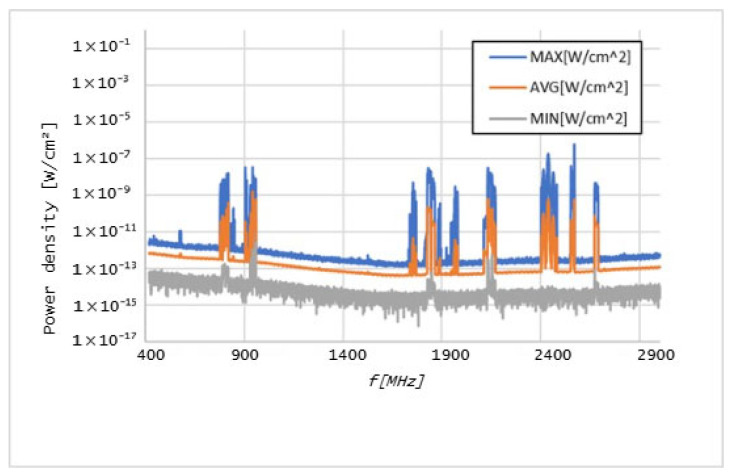
Graph of measurements at the entrance to the outpatient consultations.

**Figure 8 sensors-25-04933-f008:**
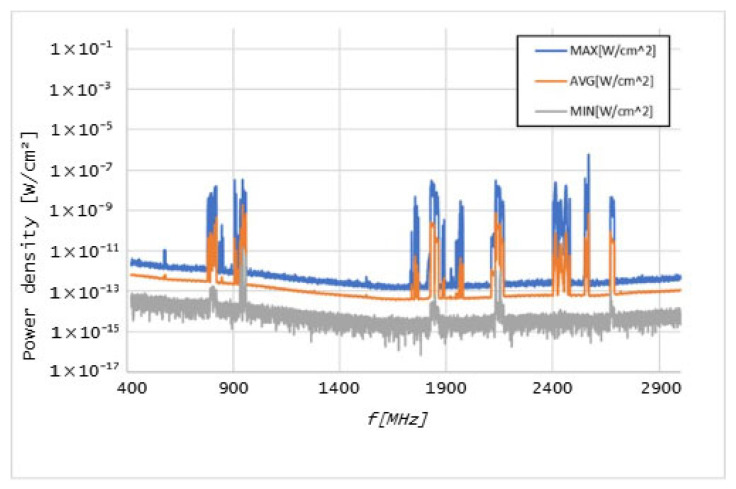
Graph of the measurements undertaken at point 17.

**Figure 9 sensors-25-04933-f009:**
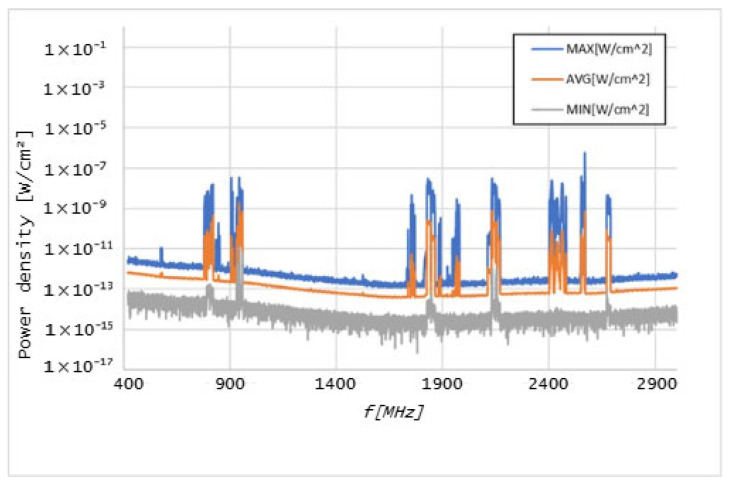
Graph of the measurements undertaken at point 19.

**Figure 10 sensors-25-04933-f010:**
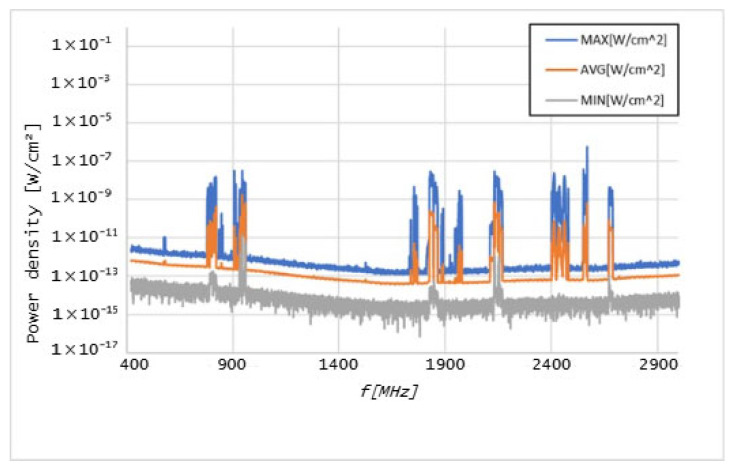
Graph of the measurements undertaken at point 21.

**Figure 11 sensors-25-04933-f011:**
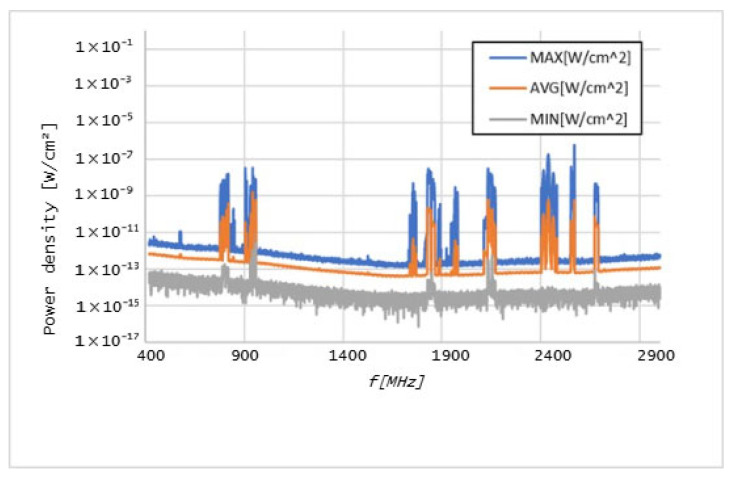
Graph of the measurements undertaken at point 23.

**Figure 12 sensors-25-04933-f012:**
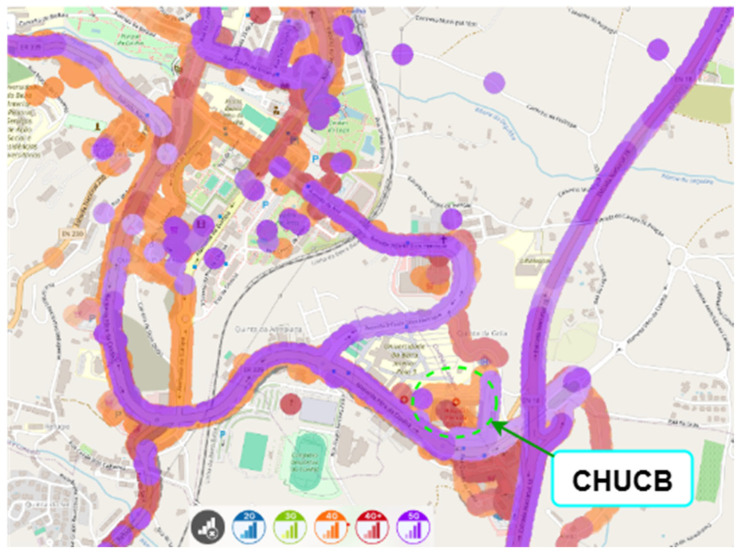
Outdoor cellular coverage map for the MEO operator in the vicinity of CHUCB. Colours represent different mobile network technologies: blue—2G, green—3G, orange—4G, red—4G+, and purple—5G. Source: nPerf platform, July 2025.

**Figure 13 sensors-25-04933-f013:**
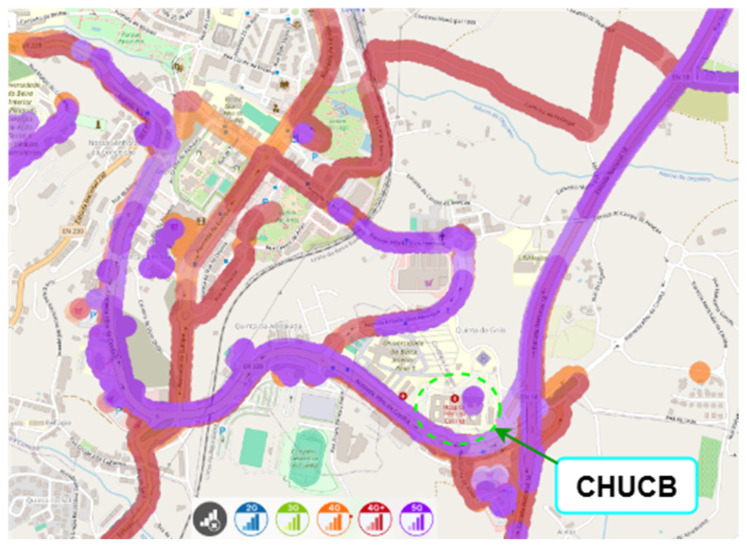
Outdoor cellular coverage map for the NOS operator in the vicinity of CHUCB. Colours represent different mobile network technologies: blue—2G, green—3G, orange—4G, red—4G+, and purple—5G. Source: nPerf platform, July 2025.

**Figure 14 sensors-25-04933-f014:**
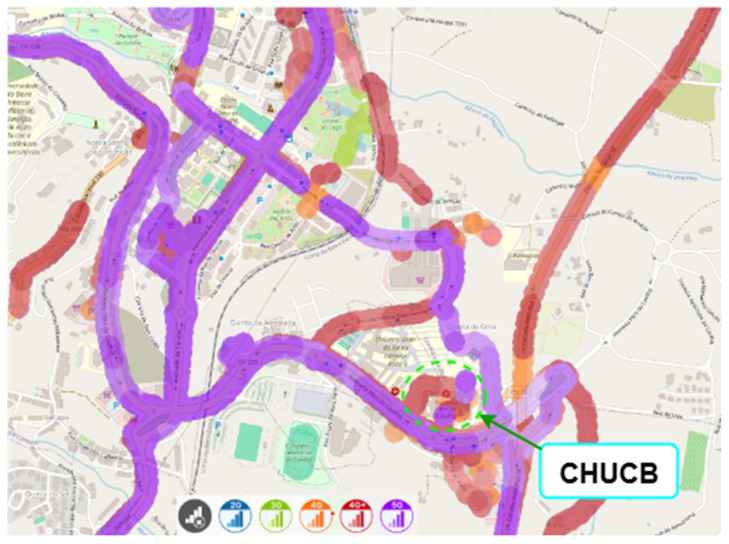
Outdoor cellular coverage map for the Vodafone operator in the vicinity of CHUCB. Colours represent different mobile network technologies: blue—2G, green—3G, orange—4G, red—4G+, and purple—5G. Source: nPerf platform, July 2025.

**Figure 15 sensors-25-04933-f015:**
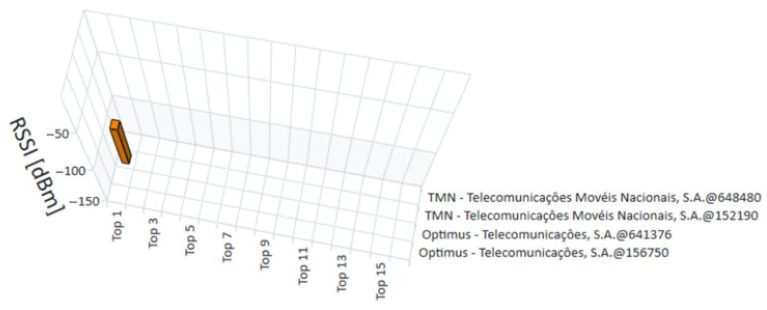
RRSI graph for 5G NR measurements at point 17.

**Figure 16 sensors-25-04933-f016:**
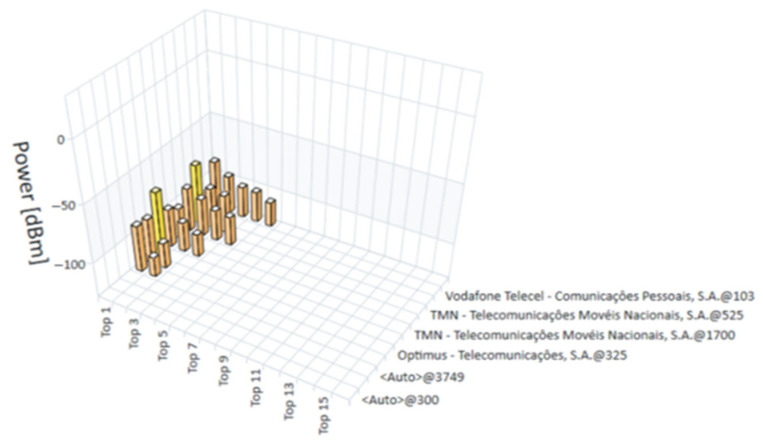
Graph of the received power for LTE measurements at point 17.

**Figure 17 sensors-25-04933-f017:**
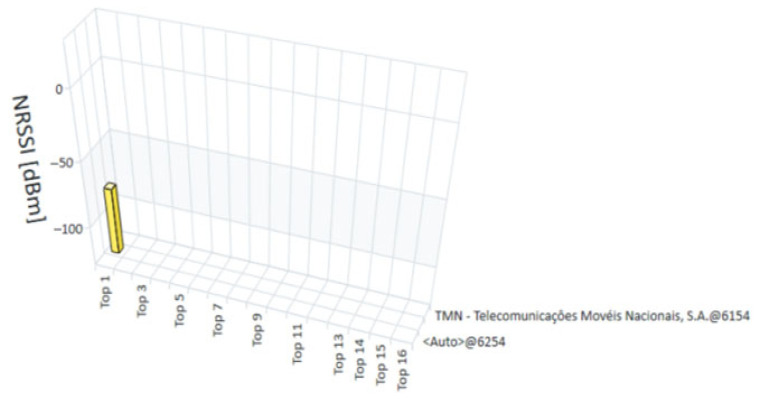
NRSSI Graph for NB-IoT measurements at point 17.

**Figure 18 sensors-25-04933-f018:**
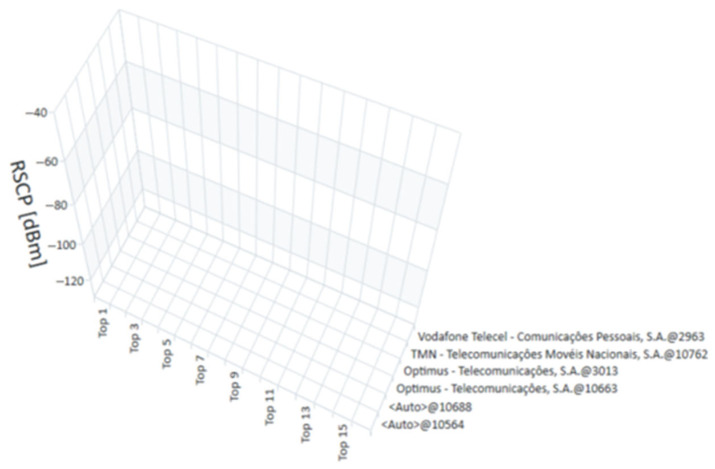
RSCP Graph for UMTS measurements at point 17.

**Figure 19 sensors-25-04933-f019:**
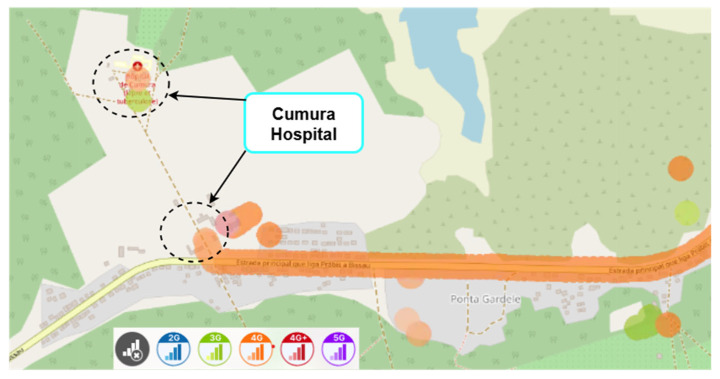
Mobile network coverage in the Cumura region (MTN operator), Guinea-Bissau. Colours represent different mobile network technologies: blue—2G, green—3G, orange—4G, red—4G+, and purple—5G. Source: nPerf platform, July 2025.

**Figure 20 sensors-25-04933-f020:**
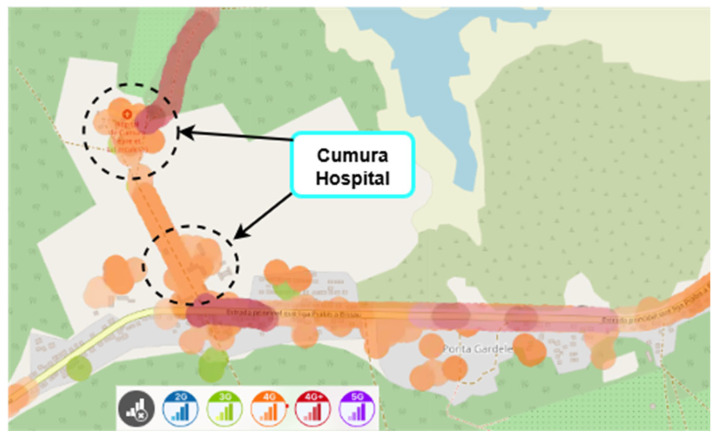
Mobile network coverage in the Cumura region (Orange operator), Guinea-Bissau. Colours represent different mobile network technologies: blue—2G, green—3G, orange—4G, red—4G+, and purple—5G. Source: nPerf platform, July 2025.

**Table 1 sensors-25-04933-t001:** Table with the designation of the measurement points.

Point/Location	Narda SRM-3006	Scanner R&S^®^TSME6
Front door	325	45 s
13	328	1:18 min
17	332	2:10 min
19	333	2:32 min
21	334	2:47 min
23	337	3:05 min

**Table 2 sensors-25-04933-t002:** Frequency bands of the Portuguese operators, based on the information provided by ANACOM.

Operator	Frequency Interval—Downlink [MHz] https://www.anacom.pt/render.jsp?categoryId=382989 (accessed on 4 February 2022)		Frequency Band [MHz]	Technology
MEO	791.0	801.0	800	Technological Neutrality
MEO	950.9	958.9	900	GSM/UMTS/WIMAX/LTE
MEO	1845.0	1865.0	1800	GSM/UMTS/WIMAX/LTE
MEO	2149.9	2169.7	2100	UMTS
MEO	2670.0	2690.0	2600	Technological Neutrality
NOS	811.0	821.0	800	Technological Neutrality
NOS	943.1	950.9	900	GSM/UMTS/WIMAX/LTE
NOS	1825.0	1845.0	1800	GSM/UMTS/WIMAX/LTE
NOS	2130.1	2144.9	2100	UMTS
NOS	2650.0	2670.0	2600	Technological Neutrality
Vodafone	801.0	811.0	800	Technological Neutrality
Vodafone	930.0	935.0	900	GSM/UMTS/WIMAX/LTE
Vodafone	935.1	943.1	900	GSM/UMTS/WIMAX/LTE
Vodafone	1805.0	1825.0	1800	GSM/UMTS/WIMAX/LTE
Vodafone	2110.3	2130.1	2100	UMTS
Vodafone	2570.0	2595.0	2600	Technological Neutrality
Vodafone	2630.0	2650.0	2600	Technological Neutrality

**Table 3 sensors-25-04933-t003:** Thresholds for each wireless technology and signal quality metric, expressed in dBm. The thresholds in Table 3 are based on 3GPP standards (TS 36.133; TS 25.133), R&S^®^TSME6 equipment documentation, and Rohde & Schwarz guidelines for 5G NR, LTE, and NB-IoT network analysis.

Technology	Metric	Unit	Typical Thresholds	Interpretation
5G NR	RSSI	dBm	>−85 · −85/−100 · ≤−100	Good · Fair · Poor
LTE	Power	dBm	>−85 · −85/−95 · ≤−95	Good · Fair · Poor
NB-IoT	NRSSI	dBm	>−90 · −90/−105 · ≤−105	Good · Fair · Poor
UMTS	RSCP	dBm	>−85 · −85/−95 · ≤−95	Good · Fair · Poor

**Table 4 sensors-25-04933-t004:** Summary of the best received power values for each operator at different measurement points for different technologies. (RSSI, Power, NRSSI, RSCP) for 5G NR, LTE, NB-IoT, and UMTS, respectively, highlighting the operator with the best coverage at each measurement point.

Measurement Points	Technology	Operator		
		MEO	NOS	Vodafone
13	5G NR (RSSI)	−100.22		
	LTE (Power)		−78.23	
	NB-IoT (NRSSI)	−104.34		
	UMTS (RSCP)		−89.2	
17	5G NR (RSSI)	−100.21		
	LTE (Power)		−74.02	
	NB-IoT (NRSSI)			
	UMTS (RSCP)			
19	5G NR (RSSI)		−100.52	
	LTE (Power)	−78.43		
	NB-IoT (NRSSI)	−92.15		
	UMTS (RSCP)		−95.4	
21	5G NR (RSSI)	−97.70		
	LTE (Power)	−83.11		
	NB-IoT (NRSSI)	−98.22		
	UMTS (RSCP)			
23	5G NR (RSSI)	−97.70		
	LTE (Power)	−80.88		
	NB-IoT (NRSSI)	−98.74		
	UMTS (RSCP)	−102.1		
Entrance to the outpatient consultations	5G NR (RSSI)		−94.5	
	LTE (Power)	−98		
	NB-IoT (NRSSI)	−100.31		
	UMTS (RSCP)	−98.6		

## Data Availability

The original contributions presented in this study are included in the article. Further inquiries can be directed to the corresponding author(s).

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
