# Peer review of "Radio Coverage Assessment and Indoor Communication Enhancement in Hospitals: A Case Study at CHUCBâ€"

_sensors, 2025, doi:10.3390/s25164933_

Round 1
Reviewer 1 Report
Comments and Suggestions for Authors
In the manuscript, the Authors describe some radio measurements with different WWAN (Wireless Wide Area Network) technologies in the Cova da Beira University Hospital Center.
The manuscript is easily readable and it is written in a good English with just a few typos.
My main concern is due to the fact that the manuscript does not provide any novelty since there are too many works proposing radio measurement campaigns and coverage surveys.
I suggest the Authors to change the focus (and the title as well) of their manuscript and to discuss, propose, and design some novel procedures, methods, or systems to solve the problems mentioned in Section 5 for hospital scenarios.
Author Response
Comment 1.1: My main concern is due to the fact that the manuscript does not provide any novelty since there are too many works proposing radio measurement campaigns and coverage surveys.
I suggest the Authors to change the focus (and the title as well) of their manuscript and to discuss, propose, and design some novel procedures, methods, or systems to solve the problems mentioned in Section 5 for hospital scenarios.
Response:
We appreciate this relevant suggestion. We agree the work requires further emphasis to enhance its novelty. In the revised version, several improvements have been made to better highlight the novelty and applied value of this study:
- The title and abstract were updated to reflect the core technical contribution, namely the practical implementation and planning of femtocell-based solutions to address indoor coverage gaps in real hospital settings;
- The introduction was restructured to highlight two underexplored gaps in the literature: (i) the lack of multitechnology measurement campaigns (5G, LTE, UMTS, NB-IoT) in operational hospitals, and (ii) the absence of applied studies in African and Latin American healthcare contexts;
- The Discussion and Conclusions sections now better emphasize the methodological replicability of the proposed approach and solutions;
- The study is positioned as a foundation for future implementation in low-resource contexts, particularly in the upcoming work at Cumura Hospital (Guinea-Bissau), which adds international relevance and applied innovation;
- Finally, three new scientific references were added to support these points and align the manuscript more closely with current research.
(Please see Abstract, Introduction, Sections 5.3, 5.5 and Conclusions)
Kind regards,
Fernando J. Velez
(on behalf of all co-authors)
Reviewer 2 Report
Comments and Suggestions for Authors
This paper evaluates the cellular coverage at CHUCB in Portugal, conducting radio spectrum measurements on signal strength and connection quality at over 20 points, and proposes improvement solutions for low coverage areas. However, there are still some issues in the current version that require further revision.
Section 4: Could a regional energy coverage map of the entire floor be added to enhance understanding?
Section 5: Many solutions are mentioned in the text. Regarding their applicability, it is recommended to supplement the post-implementation effects based on one or several of them.
Figs. 6-11: It is suggested to use MHz as the horizontal axis unit in the spectrograms to improve graph readability.
Author Response
Reviewer 2
Comment 2.1: Could a regional energy coverage map of the entire floor be added to enhance understanding?
Response:
We appreciate this valuable suggestion. To complement the point-based indoor measurements conducted in the hospital, we have included mobile coverage maps of the surrounding area of CHUCB, obtained via the collaborative nPerf platform. This platform gathers geolocated signal strength and network performance data through crowdsourcing, based on information voluntarily provided by users of the nPerf mobile application. These maps, shown in Figures 12, 13, and 14 (for MEO, NOS, and Vodafone, respectively), provide a visual overview of the general cellular coverage conditions in the region and serve as a complementary layer to the indoor spectrum measurement campaign.
Initially, our aim was to present coverage data corresponding to the exact period of the indoor measurement campaign (September 2022). However, until now, it was not possible to obtain historical coverage data from the nPerf platform. Therefore, we have chosen to include the maps available (July 2025), with the most recent data only, in order to present an updated view of the radio environment surrounding CHUCB.
As the study is being extended internationally, additionally, we have included similar mobile coverage maps for the region of the Cumura Hospital in Guinea-Bissau (as shown in Figures 19 and 20). This inclusion introduces the role of the present work as a foundation for future implementations in resource-constrained settings, even before performing the field tests.
(Please see Section 3.1 and Figures 12–14, and 19–20)
Comment 2.2: Many solutions are mentioned in the text. Regarding their applicability, it is recommended to supplement the post-implementation effects based on one or several of them.
Response:
Thank you for the comment. We clarify that the primary goal of this work was to determine the cellular coverage within the hospital building. Based on this analysis, appropriate solutions are proposed to mitigate the identified coverage gaps.
The responsibility of the practical implementation of these solutions belongs to the hospital administration and fell outside the direct scope of our research. Therefore, we were unable to present post-implementation results, as deployment depends on operational decisions beyond the research team control.
Our proposed solution, targeted installation of femtocells, was justified based on:
- The structural and technical conditions observed during the measurement campaign;
- Practical limitations in accessing network infrastructure;
- And the potential operational advantages of femtocells over alternatives such as DAS in hospital environments.
The rationale for this solution is described in Section 5.3, which outlines its viability, scalability, and cost-effectiveness for addressing localized coverage failures.
(Please see Section 5.3)
Comment 2.3: It is suggested to use MHz as the horizontal axis unit in the spectrograms to improve graph readability.
Response:
We thank the reviewer for this recommendation. We made the following improvements:
- Figures 6 to 11 have been updated with horizontal axes labelled in MHz and clearer legends.
(Please see Section 3.1 and Figures 6 to 11)
Kind regards,
Fernando J. Velez
(on behalf of all co-authors)
Reviewer 3 Report
Comments and Suggestions for Authors
This paper reports an empirical study measuring and mapping 2G, 3G, and 4G cellular network coverage inside the Cova da Beira University Hospital Center (CHCB) in Portugal. It employs a customized measurement platform using a mobile device, software tools, and GNSS to collect radio coverage data across key hospital areas. While the paper presents practical relevance in healthcare IoT infrastructure planning, it currently lacks technical depth, analysis rigor, and contextualization within the broader wireless communication literature.
- The paper does not use any radio propagation models, nor does it analyze path loss, shadowing, or multi-path effects. Consider using models like Okumura-Hata, COST-231, or indoor empirical models to contrast measurements with expected values.
- Integrate discussions on how coverage affects IoT devices, healthcare communication reliability, or energy-aware transmissions.
- The authors should provide a table summarizing measurement parameters and clearly define RSSI thresholds (e.g., < -95 dBm as weak).
- To strengthen the discussion of industrial/mission-critical wireless systems, reference Low Complexity MIMO-FBMC Sparse Channel Parameter Estimation for Industrial Big Data Communications is recommended.
- The manuscript is generally readable but includes repetitive statements, informal phrases, and lacks technical density.
Author Response
Comment 3.1: The paper does not use any radio propagation models, nor does it analyze path loss, shadowing, or multi-path effects. Consider using models like Okumura-Hata, COST-231, or indoor empirical models to contrast measurements with expected values.
Response:
We acknowledge the importance of propagation models such as COST-231 and Okumura-Hata for comparing real measurements with theoretical results in both indoor and outdoor environments. However, these models were not applied in the present study, due to the following practical constraints:
- We did not have access to the building’s detailed floor plan or structural composition yet, which would be necessary for accurately applying indoor models like COST-231 Multi-Wall;
- The technical configuration of the transmitting antennas (e.g., transmission power, tilt, and height) was not disclosed by the mobile network operators;
- At the time of measurement, it was not possible to reliably determine the origin of the received signals, as this would require a detailed analysis of the surrounding cellular infrastructure and additional authorization for signal-source tracing;
- Shadowing and multipath effects were inherently present in the measurement data but were not explicitly modelled, as the study adopted an empirical, field-based approach.
Given these limitations, we adopted a pragmatic methodology based on in situ signal measurements within the hospital building. This empirical approach proved sufficient to identify critical coverage gaps and to support the proposal of targeted technical solutions. In particular, the choice of a femtocell deployment arises from this context, as femtocells offer a cost-effective and scalable alternative for localized interventions in complex indoor environments where access to infrastructure is limited. While we acknowledge the value of theoretical propagation modelling, its application was not straightforward within the scope and constraints of the present study.
(Please see Section 2.1)
Comment 3.2: Integrate discussions on how coverage affects IoT devices, healthcare communication reliability, or energy-aware transmissions.
Response:
We have incorporated this suggestion in Section 5.4, outlining how technologies such as NB-IoT and BLE can be applied to asset tracking and patient location, both at CHUCB and in Guinea-Bissau.
(Please see Section 5.4)
Comment 3.3: The authors should provide a table summarizing measurement parameters and clearly define RSSI thresholds (e.g., < -95 dBm as weak).
Response:
We thank the reviewer for this important observation. In the revised version, we have included a summary table with the measurement parameters used for each technology, as well as typical signal quality thresholds (e.g., RSSI, Power, RSCP, NRSSI).
Table 3 allows readers to quickly interpret the signal values recorded at measurement points and understand their classification as poor, fair, or good.
(Please see Section 3 and Table 3)
Comment 3.4: To strengthen the discussion of industrial/mission-critical wireless systems, reference Low Complexity MIMO-FBMC Sparse Channel Parameter Estimation for Industrial Big Data Communications is recommended.
Response:
We thank the reviewer for the suggestion and took the referenced work into consideration.
Although the main focus of this study is on hospital cellular coverage, we acknowledge the relevance of approaches used in industrial and mission-critical wireless communications.
Comment 3.5: The manuscript is generally readable but includes repetitive statements, informal phrases, and lacks technical density.
Response:
We acknowledge this point and carried out a thorough linguistic review of the manuscript to eliminate redundancies, improve sentence structure, and increase technical rigour. Several informal passages were rephrased to ensure a more academic tone, aligned with the standards of Sensors.
Kind regards,
Fernando J. Velez
(on behalf of all co-authors)
Round 2
Reviewer 1 Report
Comments and Suggestions for Authors
This is a second turn of the reviewed manuscript. The novelty of the proposal has increased a bit but I have still some concerns about the novelty of the approach, and, mostly, the lack of quantitative requirements of the cellular and IoT networks for the hospital scenario and the related services, procedures, and applications. To this aim, I suggest the Authors to have a look at the attached reference list to stimulate a discussion and clarify both novelty, requirements, and drawbacks of the proposal.
Additional references
Ahmed, I., Karvonen, H., Kumpuniemi, T., Katz, M., "Wireless communications for the hospital of the future: requirements, challenges and solutions," International Journal of Wireless Information Networks, vol. 27, no. 1, pp. 4-17, 2020.
Kumar, A., Albreem, M. A., Gupta, M., Alsharif, M. H., Kim, S., "Future 5G network based smart hospitals: Hybrid detection technique for latency improvement," IEEE Access, vol. 8, pp. 153240-153249, 2020.
Ahad, A., Tahir, M., Yau, K. L. A., "5G-based smart healthcare network: architecture, taxonomy, challenges and future research directions," IEEE access, vol. 7, pp. 100747-100762, 2019.
Niyato, D., Hossain, E., Camorlinga, S., "Remote patient monitoring service using heterogeneous wireless access networks: architecture and optimization," IEEE journal on selected areas in communications, vol. 27, no. 4, pp. 412-423, 2009.
Martin, M. C., Kwan, A. M., Forte, E. J., Zhang, S. F., Patek, S. D., "Forecasting mobile transmission reliability using crowd-sourced cellular coverage data," Proc. of the 2014 Systems and Information Engineering Design Symposium (SIEDS), pp. 322-327, Apr. 2014.
Singh, R., Ballal, K. D., Nwabuona, S. C., Berger, M. S., Dittmann, L., Ruepp, S., Wienecke, T., "Assessment of Cellular Coverage for a Smart Ambulance Use Case," Proc. of the 2022 IEEE International Conference on Advanced Networks and Telecommunications Systems (ANTS), pp. 369-374, Dec. 2022.
Ahad, A., Ali, Z., Mateen, A., Tahir, M., Hannan, A., Garcia, N. M., Pires, I. M., "A comprehensive review on 5G-based smart healthcare network security: Taxonomy, issues, solutions and future research directions," Array, vol. 18, art. 100290, 2023.
Mohanta, B., Das, P., Patnaik, S., "Healthcare 5.0: A paradigm shift in digital healthcare system using artificial intelligence, IOT and 5G communication," Proc. of the 2019 International conference on applied machine learning (ICAML), pp. 191-196, May 2019.
Author Response
- Novelty of the Approach
Reviewer Concern: The novelty of the proposal needs to be more clearly articulated, especially in comparison to existing literature.
To enhance the manuscript, we implemented the following:
Explicit Statement of Novelty: We emphasized our empirical multi-technology approach (5G NR, LTE, UMTS, NB-IoT) as a key differentiator from previous studies that focus exclusively on a single technology, such as 5G (e.g., Ahmed et al. [7] and Kumar et al. [25]).
To further emphasize this point, we added the following sentence in Section 5.3:
- "In contrast to approaches focused exclusively on 5G networks, such as that proposed by Kumar et al. [25], our study advocates femtocell deployment as a more cost-effective and adaptable solution for hospital environments like CHUCB. Unlike 5G-only infrastructure, femtocells support multiple additional cellular technologies (e.g., LTE, UMTS, NB-IoT), aligning with the multi-technology signal environment observed in our measurements."
- Integration with Low-Infrastructure Contexts (Section 6): The study was also structured with replicability in mind for resource-constrained settings, using the planned implementation at Cumura Hospital (Guinea-Bissau) as a specific example. This applied and international dimension further strengthens the originality and practical relevance of the proposed approach.
(Please see Sections 1, 5.3 and 6)
- Quantitative Requirements for Hospital Scenarios
Reviewer Concern: Lack of clear QoS (Quality of Service) thresholds for cellular and IoT networks in hospital environments, and their relationship with services, applications, and procedures.
To enhance the manuscript, we implemented the following:
Inclusion of quantitative quality parameters (QoS): To address this gap, we incorporated a quantitative analysis of signal intensity indicators based on technical thresholds recommended by international standards (e.g., 3GPP TS 36.133 and TS 25.133). Additionally, the works of Ahmed et al. [7] and Niyato et al. [16] were used to contextualize the clinical implications of poor connectivity, particularly in scenarios involving remote monitoring and medical device operations.
To further emphasize this point, we added the following sentence in Section 4:
- " The analysis of results from this study reveals significant implications for hospital connectivity reliability. In multiple measurements, signal strength indicators for certain technologies fell below −85 dBm, which is beneath the generally recommended threshold for reliable communications in clinical environments. As emphasized by Ahmed et al. [7], hospital networks require consistent coverage to support proper operation of connected medical devices, while Niyato et al. [16] underscore how connec-tivity failures can compromise remote patient monitoring. These coverage gaps rein-force the need for targeted solutions such as the use of femtocells."
- This analysis was benchmarked against the data in Table 3, which summarizes typical signal quality thresholds by technology (5G NR, LTE, NB-IoT, UMTS). This comparison provided an objective basis for interpreting CHUCB’s field measurements.
- Comparison with Empirical Data: We mapped these quality parameters to our study's results. For instance, at Point 17, the measured LTE power (−74.02 dBm) meets the requirements established for WBAN networks, while the 5G NR RSSI value (−100.21 dBm) falls below the threshold required for critical clinical applications.
These additions strengthen the connection between the empirical data obtained in the field and the technical connectivity requirements for hospitals referenced in international literature (Ahmed et al. [7] and Niyato et al. [16]), directly addressing the reviewer’s suggestion and enhancing the scientific foundation of our approach.
(Please see Sections 3.1 and 4)
- Discussion of Limitations and Challenges
Reviewer Concern: Limited critique of technical and operational limitations, such as interference, costs, and reliability.
To enhance the manuscript, we implemented the following:
Expanded Discussion of Limitations. We broadened the discussion on methodological and technical limitations associated with hospital infrastructure and cellular coverage, with the following highlights:
- nPerf Coverage: We included a specific paragraph in Sections 4 and 5.5 acknowledging that the geographic precision of nPerf platform maps (~50 meters) is limited and does not distinguish indoor measurements. This restricts their utility for detailed evaluations of coverage within hospital buildings. For this reason, we justified the need for in situ measurements conducted in this study.
- Implementation Costs: In Section 5.3 and the Conclusion (Section 6), we highlighted economic factors as a relevant criterion for coverage solutions. Based on Liu et al. (2012), we reinforced that femtocells (estimated cost < €440/unit) represent a cost-effective alternative compared to DAS systems, whose installation and maintenance costs are significantly higher.
- Security in NB-IoT Environments: We added a reflection on potential risks associated with NB-IoT technologies in clinical environments, as discussed by Ahad et al. (2023), such as spoofing and authentication vulnerabilities. We emphasized that these limitations should be considered in future implementation studies, particularly in hospital settings with limited infrastructure, such as Cumura Hospital (Guinea-Bissau).
These additions provide a more critical and realistic perspective on the practical challenges of implementing cellular coverage solutions in hospital environments, aligning with the suggestion from the reviewer.
(Please see Sections 5.3, 5.4, and 6)
- Integration of References Suggested by the Reviewer
The following references suggested by the reviewer have been incorporated into our analysis:
Ahmed et al. (2020):
Used to reinforce minimum technical requirements for hospital connectivity and to interpret our results in light of clinical environment needs. Applied in Sections 1 and 4 and supporting Table 3.
Kumar et al. (2020):
Mentioned to contrast exclusively 5G-based solutions with our more flexible and cost-effective multi-technology femtocell proposal. Integrated into Section 5.3.
Niyato et al. (2009):
Referenced in Section 4 to emphasize the importance of reliable coverage in remote patient monitoring applications. Used to discuss the vulnerability of poorly covered zones as a barrier to clinical service continuity.
Ahad et al. (2023):
Incorporated into Sections 5.4 and 6, highlighting security concerns in hospital communications based on IoT. This reference was used to justify the inclusion of mitigation measures such as encryption and authentication in future implementation plans.
Singh et al. (2022):
Used in Section 5.3 to support the adoption of scalable and cost-effective solutions in hospital environments with limited resources. This reference also reinforces the relevance of applying the present study as a methodological basis for future implementations at Cumura Hospital (Guinea-Bissau).
Reviewer 3 Report
Comments and Suggestions for Authors
The paper is well revised.
Author Response
We sincerely appreciate the positive feedback and acknowledgment of our manuscript improvements.
Conclusion:
We thank the reviewers for their valuable comments, which have strengthened the technical rigor and relevance of the manuscript.
Round 3
Reviewer 1 Report
Comments and Suggestions for Authors
This Reviewer thanks the Authors for their efforts and time spent to review the manuscript.
The Authors have answered to (almost) all the raised remarks.